# A copula based topology preserving graph convolution network for clustering of single-cell RNA-seq data

**Snehalika Lall**[1], **Sumanta Ray**[2,3]*, **Sanghamitra Bandyopadhyay**[1]*

**1** Machine Intelligence Unit, Indian Statistical Institute, Kolkata, India, **2** Department of Computer Science and Engineering, Aliah University, Kolkata, India, **3** Health Analytics Network, Pittsburgh, Pennsylvania, United States of America

* sumanta.ray@aliah.ac.in, sray@healthanalytics.net (SR); sanghami@isical.ac.in (SB)

**Data Availability Statement:** Data and Code are available in the github page. All the datasets can be downloaded from the public repository. Baron dataset can be downloaded from https://hemberg-

## Abstract

Annotation of cells in single-cell clustering requires a homogeneous grouping of cell populations. There are various issues in single cell sequencing that effect homogeneous grouping (clustering) of cells, such as small amount of starting RNA, limited per-cell sequenced reads, cell-to-cell variability due to cell-cycle, cellular morphology, and variable reagent concentrations. Moreover, single cell data is susceptible to technical noise, which affects the quality of genes (or features) selected/extracted prior to clustering.

Here we introduce sc-CGconv (**c**opula based **g**raph **conv**olution network for **s**ingle **c**lustering), a stepwise robust unsupervised feature extraction and clustering approach that formulates and aggregates cell–cell relationships using copula correlation (Ccor), followed by a graph convolution network based clustering approach. sc-CGconv formulates a cell-cell graph using *Ccor* that is learned by a graph-based artificial intelligence model, graph convolution network. The learned representation (low dimensional embedding) is utilized for cell clustering. sc-CGconv features the following advantages. a. sc-CGconv works with substantially smaller sample sizes to identify homogeneous clusters. b. sc-CGconv can model the expression co-variability of a large number of genes, thereby outperforming state-of-the-art gene selection/extraction methods for clustering. c. sc-CGconv preserves the cell-to-cell variability within the selected gene set by constructing a cell-cell graph through copula correlation measure. d. sc-CGconv provides a topology-preserving embedding of cells in low dimensional space.

## Author summary

One of the important aspects of single cell downstream analysis is to classify cells into subpopulations. This immediately leads to clustering of cells into homogeneous groups, which faces lots of issues due to (i) small amount of starting RNA, (ii) cell-to-cell variability, (iii) technical noise incorporated within the single cell sequencing technology, and (iv) unavailability of discriminating selected/extracted genes (features) in the preprocessing step of downstream analysis. We proposed sc-CGconv, stepwise feature extraction and

lab.github.io/scRNA.seq.datasets/human/pancreas/
Klein dataset can be downloaded from https://
hemberg-lab.github.io/scRNA.seq.datasets/mouse/
esc/ Melanoma data is available in GEO under
accession no. GSE72056. PBMC can be
downloaded from https://support.10xgenomics.
com/single-cell-geneexpression/datasets.

**Funding:** SB would like to acknowledge support
from J.C. Bose Fellowship (SB/S1/JCB- 033/2016
to S.B.) by the DST, Govt. of India; SyMeC Project
grant (BT/Med-II/NIBMG/SyMeC/2014/Vol. II)
given to the Indian Statistical Institute by the
Department of Biotechnology (DBT), Govt. of India.
Govt. of India; Inspire DST Project. SR would like
to acknowledge support from SERB TARE grant
(File No. TAR/2021/000072) by the DST, Govt. of
India. The funders had no role in study design, data
collection and analysis, decision to publish, or
preparation of the manuscript.

**Competing interests:** The authors have declared
that no competing interests exist.

clustering framework, which leverage landmark advantage of copula and graph convolution network in single-cell analysis domain. sc-CGconv outperforms the state-of-the-art feature selection/extraction methods in the preprocessing steps, performs well with small sample size data, can preserve the cell-to-cell variability within the extracted features, provides a topology-preserving embedding of cells in low dimensional space. sc-CGconv therefore successfully addresses the above-mentioned key challenges.

This is a *PLOS Computational Biology* Methods paper.

## Introduction

Recent developments of single cell RNA-seq (scRNA-seq) technology made it possible to generate a huge volume of data allowing the researcher to measure and quantify RNA levels on large scales [1]. This has led to a greater understanding of the heterogeneity of cell population, disease states, cell types, developmental lineages, and many more.

A fundamental goal of scRNA-seq data analysis is cell type detection [2, 3]. The most immediate and standard approach performs clustering to group the cells, which are later labeled with specific type [4, 5]. This provides an unsupervised method of grouping similar cells into clusters that facilitate the annotation cells with specific types present in the large population of scRNA-seq data [3, 6, 7].

The standard pipeline of downstream analysis of scRNA-seq data starts from the processing of the raw count matrix, and goes through the following steps [8, 9]: i) normalization (and quality control) of the raw count matrix ii) gene selection, and cell filtering iii) dimensionality reduction, iv) unsupervised clustering of cells into groups (or clusters) and v) annotation of cells by assigning labels to each cluster. Clustering of cells is not a distinct process in the downstream analysis, instead is a combination of several steps starting from step-(i) to step-(iv). Each step has an immense impact on the cell clustering process. A good clustering (or classifying cell samples) can be ensured by the following characteristics of features obtained from the step-(iii): the features should contain information about the biology of the system, should not have features containing random noise, and should preserve the structure of data while reducing the size as much as possible.

Although there are a plethora of methods [1, 10–14] available for performing each task within the pipeline, the standard approaches consider a common sequence of steps for the preprocessing of scRNA-seq data [15, 16]. This includes normalization by scaling of sample-specific size factors, log transformation, and gene selection by using the coefficient of variation (highly variable genes [5, 17]) or by using average expression level (highly expressed genes). Scanpy used several dispersion based methods [1, 18] for selecting highly variable genes (HVG). In Seurat package [19], standardized variance is calculated from the normalized data to find out the HVGs. Alternatively, some methods exist for gene selection, such as GLM-PCA [15] selects genes by ranking genes using deviance, M3Drop [12] selects genes leveraging the dropouts effects in the scRNA-seq data. After gene selection and dimensionality reduction, most of the methods for single cell clustering followed a Louvain/Leiden based graph clustering method (Scanpy and Seurat). The standard approaches for gene selection (or feature extraction) fail to produce a stable and predictive feature set for higher dimension scRNA-seq data [20]. Moreover, the existing approaches overlook the cellular heterogeneity and patterns across transcriptional landscapes, which ultimately affects cell clustering. This motivates to go

for a robust and stable technique that can deal with the larger dimension of the single cell data, while preserving the cell-to-cell variability.

Here, we introduce sc-CGconv, stepwise feature extraction and clustering approach that leverages the Copula-based dependency measure [13] and its implication for identifying stable features from large scRNA-seq data. Notably, this approach largely and effectively masks all the aforementioned limitations associated with other feature selection and clustering approaches in unsupervised cases. sc-CGconv takes a two-step strategy: first, a structure-aware gene sampling based on Local Sensitive Hashing (LSH) is performed to obtain a sub-optimal gene set [21]. In the next step, a copula-based multivariate dependency measure is utilized to map the cells into a graph which is further learned by a graph convolution network. A robust-equitable copula correlation (Ccor) is utilized for constructing the cell-cell graph. The first step ensures preserving the cell-to-cell dependence structure within the sub-sample of genes, while the second step puts all the cells into an encompassing context retaining the dependency structure among the cells, resulting in a topology-preserving embedding of the fine-grained graph using the GCN. The latent embedding resulting from the trained GCN is utilized for clustering.

The advantages of sc-CGconv are: a new robust-equitable copula correlation (Ccor) measure for constructing cell-cell graph leveraging the scale-invariant property of Copula, and reducing the computational cost of processing large datasets due to the use of structure-aware using LSH. Furthermore, to highlight the potency of sc-CGconv over the existing methods, we compared our method with seven well-known gene selection and clustering methods of scRNA-seq data: *Gini-clust* [22], *GLM-PCA* [15], *M3Drop* [12], *Fano factor*, and *HVG* followed by clustering of Seurat V3/V4 [23], *scGeneFit* [24] followed Kmeans clustering, and SC3 clustering [4]. For the methods which are specific for gene selection (GLM-PCA, M3drop) we perform kmeans clustering after selecting the informative genes. We further carry out a stability test to prove the efficacy of sc-CGconv for producing topology preserving embedding of cells from scRNA-seq data in the presence of technical noise. The results show that sc-CGconv not only can extract the most informative and non-redundant features but is also less sensitive towards technical noise present in the scRNA-seq data.

We demonstrate in experiments that *(i)* sc-CGconv leads to a pure clustering of cells in scRNA-seq data, *(ii)* the annotation of cells is accurate for unknown test samples *(iii)* the marker genes which are identified in the annotation step have a clear capability to segregate the cell types in the scRNA-seq data, and *(iv)* sc-CGconv can handle substantially large data with utmost accuracy.

## Results

### Overview of sc-CGconv

In the following, we present the workflow of the proposed method *sc-CGconv*.

### sc-CGconv: Workflow

See Fig 1 for the workflow of our analysis pipeline. We describe all important steps in the paragraphs of this subsection.

**A. Preprocessing of raw datasets.** See Fig 1A. Raw scRNA-seq datasets are obtained from public data sources. The counts matrix $M \in \mathcal{R}^{c \times g}$, where $c$ is number of cells and $g$ represents number of genes, is normalized using a transformation method called (Linnorm) [25]. We choose cells having more than 1000 genes with non-zero expression values and choose genes having a minimum read count greater than 5 in at least 10% among all the cells. $log_2$ normalization is employed on the transformed matrix by adding one as a pseudo count.

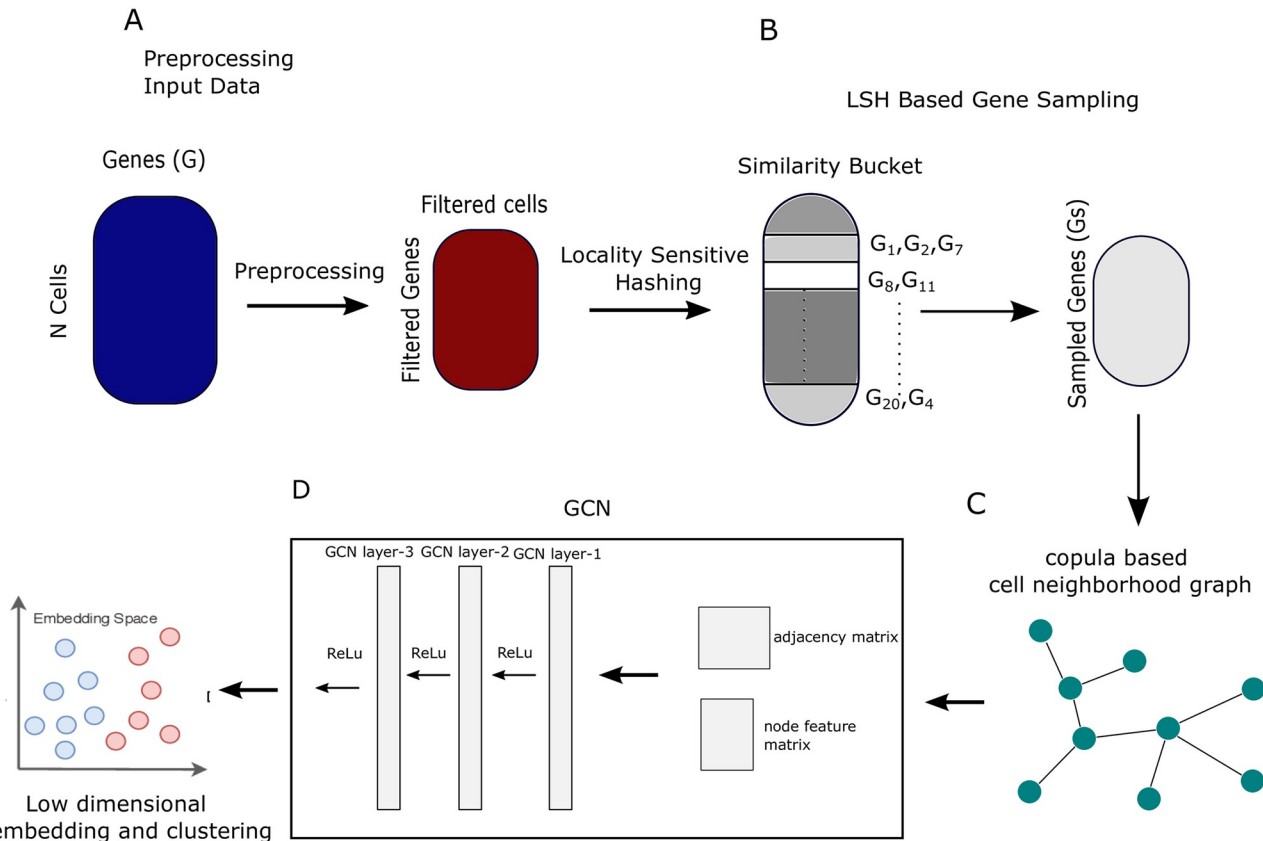

**Fig 1. Workflow of the analysis.** A. scRNA-seq count matrix are downloaded and preprocessed using linnorm. B. LSH based sampling is performed on the preprocessed data to obtain a subsample of features. C. A cell neighbourhood graph is constructed using copula correlation. D. A three layer graph convolution neural network is learned with adjacency matrix and node feature matrix as input. It aggregates information over neighbourhoods to update the representation of nodes. The final representation obtained is called graph embedding which is utilized for cell clustering.

**B. Structure-aware feature sampling using LSH.** See Fig 1B. Here, LSH is used to partition the data points (genes) of the preprocessed count matrix into different buckets. Locality Sensitive Hashing (LSH) [26–28] operates on a reduced dimension to find approximate nearest neighbors. LSH uses special locality-sensitive hash functions where the chances of similar objects hashing into the same bucket are more than the collision of dissimilar objects, [29]. By virtue of the hash function the relative distance between the data points in the input space is preserved in the output space. A $k$-nn graph is formed by searching the five nearest neighbours within the bucket for each gene. A sub-sample of genes is obtained by performing a greedy approach for selecting the genes within each bucket. It results in a subset of genes, that are considered as feature set, $F_{LSH} = \{f_s : s = 1, \cdots, m\}$ for the cell-cell graph construction in the next stage. Here the aim is to find out an important non-redundant subset of features (genes) while preserving the structure of the data.

**C. Constructing cell-cell graph using copula correlation.** See Fig 1C. A robust equitable correlation measure *Ccor* is utilized to measure the dependence between each pair of cells over the sampled transcriptome obtained from the LSH step. For each node (cell) a ranked list of nodes (cells) is generated based on the *Ccor* scores. We assume a cell pair having a larger *Ccor* value shares the most similar transcriptomic profile. Next, a k-nearest neighbour graph is constructed based on the ranked list of each node (cell).

**D. Learning low dimensional embedding from cell-cell-graph.** See Fig 1D. We employ a network embedding strategy (here: Graph Convolution Network [30]), which extracts node features from the constructed cell-cell graph. In detail, GCN has the advantage that it can utilize the power of convolution neural network to encode the relationship between samples. The graph structure (generally represented as adjacency matrix) together with the information encoded in each node is utilized in the NN. We encode the entire graph (adjacency matrix) into a fixed-size, low-dimensional latent space. Thus GCN encoder preserves the properties of all the nodes (cells) relative to their encompasses in the network. The result of this step is a feature matrix where rows refer to nodes (cells) and columns refer to the inferred network features.

## Training of graph convolution network on cell-cell graph

To train the GCN model with our datasets, we first randomly split the cell-cell graph into an 8:1:1 ratio of the train, validation, and test sets. The test edges are not included in the training set, however, we keep all the nodes of the graph in the training set. Now, we train the model using the training edges and check the performance of the trained model for recovering the removed test edges. The model is trained with 50 epochs using Adam optimizer with a learning rate of 0.001 and a dropout rate of 0.1. Adam [31] is an optimization algorithm that can be used instead of the classical stochastic gradient descent procedure to update network weights iterative based on training data. It is appropriate for problems with very sparse gradients. The rectified linear activation function (ReLU) is a piecewise linear function that will output the input directly if it is positive, otherwise, it will output zero. It has become the default activation function for most of the neural networks models because of its simplicity and better performance. The learning rate is initially chosen as 0.001 as it gives better performance in our experiments. Table 1 shows the average precision and receiver operating characteristic (ROC) score (ROC plot is given in Fig A in S1 File) for the four networks obtained from the datasets. We took the low dimensional embeddings from the output of the encoder of the trained model.

## sc-CGconv can produce topology-preserving single-cell embedding

The resulting embedding of sc-CGconv can be utilized to generate the single-cell embeddings for clustering. Here we compare sc-CGconv with three manifold learning and graph drawing algorithms such as UMAP, t-SNE, and ForceAtlas2 to quantify the quality of resulting embedding. To see how similar the topology of low-dimensional embedding (within the latent space $Z$) is to the topology of the high-dimensional space ($X$), we adopted a procedure similar to Wolf et al. [35]. Here, we define a classification setup where the ground truth is defined as a kNN graph $G_X = (V, E_X)$ fitted in the high dimensional space $X$. The edge set $E_{FC}$ which defines all possible edges is the state space of the classification problem. In this setting, the embedding

**Table 1. Performance of GCN on networks created from four datasets.** First two columns of the table shows total number of edges and number of nodes of the four networks. The rest of the columns show ROC and average precision score for validation and test edges. V. ROC and V. AP refer to validation ROC and validation average precision score, whereas T. ROC and T. AP refer to the same for test set.

| Dataset | #edges | #nodes | V. ROC | V. AP | T. ROC | T. AP |
|---|---|---|---|---|---|---|
| Baron [32] | 41876 | 8569 | 87.32 | 87.08 | 85.87 | 86.39 |
| Klein [33] | 13885 | 2717 | 84.79 | 83.21 | 83.46 | 82.81 |
| Melanoma [34] | 340875 | 68579 | 83.38 | 86.48 | 83.1 | 82.30 |
| PBMC68k [1] | 342890 | 68793 | 84.98 | 86.78 | 82.9 | 83.8 |

algorithm predicts whether an edge $e \in E_{FC}$ is an element of $E_X$. We put label '1' for the edge $e \in E_{FC}$ if $e \in E_X$, otherwise put label '0'. For each edge $e \in E_{FC}$, the embedding will put label '1' with the probability $q_e$ and put label '0' with probability $1 - q_e$. The cost function to train such a classifier is formed as a binary cross-entropy function $H(P, Q)$ or logloss, which is equivalent to the negative log-likelihood of the labels under the model. It is defined as

$$H(P, Q) = \sum_{e \in E_{FC}} \sum_{l \in 0,1} p_e log(q_e) = \sum_{e \in E_{FC}} p_e log\left(\frac{1}{q_e}\right) + (1 - p_e)log\left(\frac{1}{1 - q_e}\right) \qquad (1)$$

Now the Kullback-Leibler (KL) [36] divergence of the predicted distribution $Q$ and the reference distribution $P$ is measured as $KL(P, Q) = H(P, Q) - H(P)$, where $H(P) = -\sum_{e \in E_{FC}} p_e$, which ultimately leads to the equation

$$KL(P, Q) = \sum_{e \in E_{FC}} p_e log\left(\frac{p_e}{q_e}\right) + (1 - p_e)log\left(\frac{1 - p_e}{1 - q_e}\right). \qquad (2)$$

We measured the KL divergence between $P$ and $Q$ for t-SNE [37], UMAP [38], ForceAtlas2 [39], PHATE [40], SAUCIE [41], and the sc-CGconv. Fig 2 shows the statistics of KL measures for the different embeddings in the four used datasets.

## Comparison with State-Of-The-Art Methods

In scRNA-seq datasets, single cells are the unit of analysis, and it is crucial to correctly identify the clusters to which they belong. These reference clusters are typically based on the expression profiles of many cells. Misclassification of cells is the common issue for annotating clusters as single-cell gene expression datasets often show a high level of heterogeneity even within a given cluster. To establish the efficacy of sc-CGconv over such procedures, we have selected seven state-of-the-art methods that are widely used for gene selection and clustering of the single cell data.

Here, we compare sc-CGconv with the following seven methods I) *Gini Clust* [22]: a feature selection scheme using Gini-index followed by density-based spatial clustering of applications with noise, DBSCAN [42]. II) *GLM-PCA* [15]: a multinomial model for feature selection and dimensionality reduction using generalized principal component analysis (GLM-PCA) followed by k-means clustering III) Seurat V3/V4 [23]: a single cell clustering pipeline which selects *Highly Variable Gene (HVG)* that exhibit high cell-to-cell variation in the dataset (i.e, they are highly expressed in some cells, and lowly expressed in others) followed by Louvain clustering IV) *Fano Factor* [43], a measure of dispersion among features. Features having the maximum Fano Factor are chosen for Kmeans clustering. V) scGeneFit [24], a marker selection method that jointly optimizes cell label recovery using label-aware compressive classification, resulting in a substantially more robust and less redundant set of markers. Vi) SC3 [4], a single-cell consensus (Kmeans) clustering (SC3) tool for unsupervised clustering of scRNA-seq data. Vii) M3drop [44] which takes advantage of the prevalence of zeros (dropouts) in scRNASeq data to identify features.

The R package for Gini-Clust [22] is employed with default settings. For GLM-PCA and HVG, we consider the default settings as described in [15, 23]. For scGene-Fit, the parameters (redundancy = 0.25, and method = 'centers') are used as suggested by their github page. For SC3 we adopted the default parameters for clustering all the datasets.

sc-CGconv requires the number of iterations (*iter*) as the input parameters of LSH step. we set as *iter* as 1 for all datasets. The parameter of Clayton Copula (see section Method for details) is set as $\theta = -0.5$. This $\theta$ is set as it gives better performance in our experiments. For

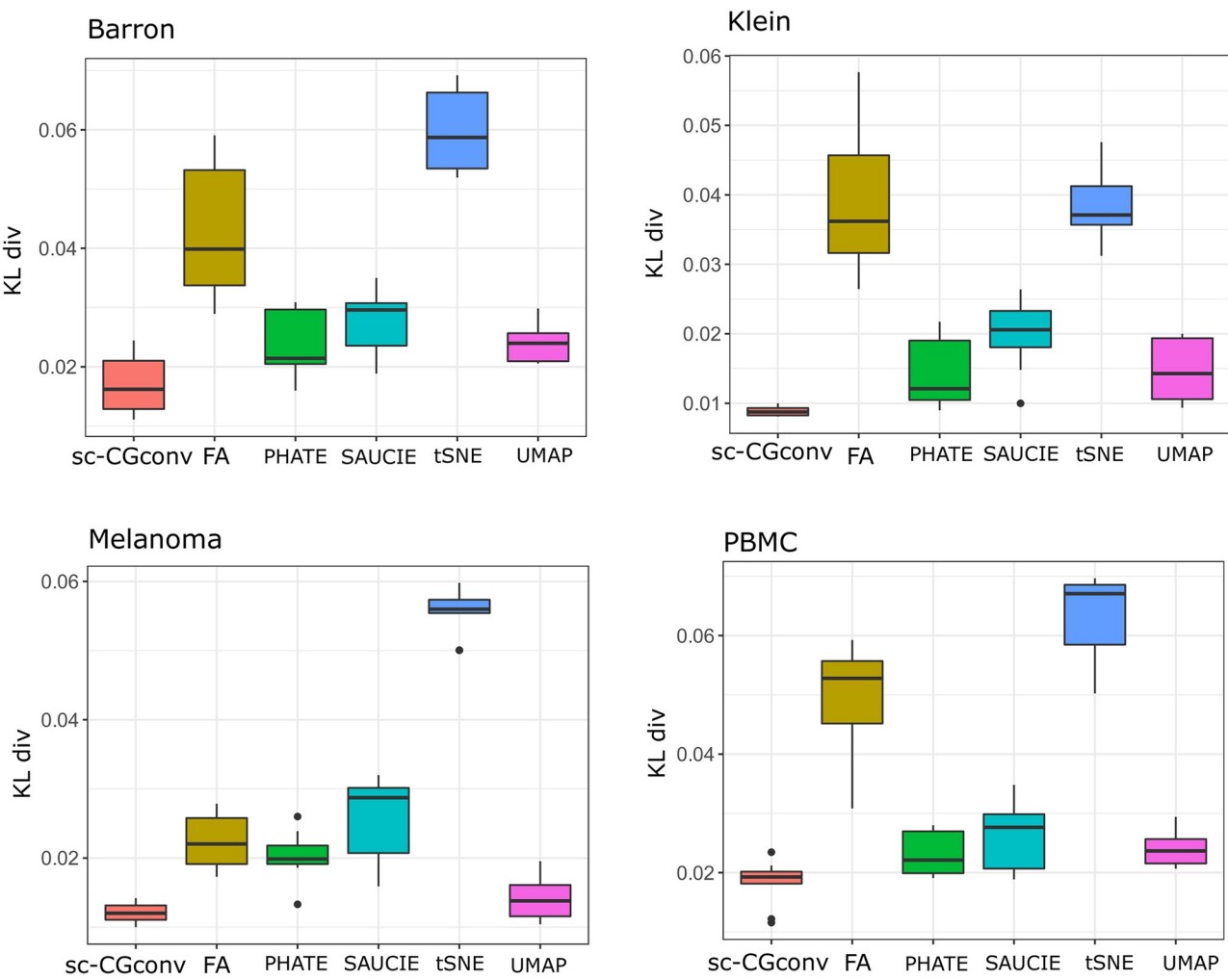

**Fig 2. Performance of different embedding algorithms on four datasets.** Kl divergence (KL div) is computed by rerunning embedding algorithms 50 times.

GCN, we used 3-layer GCN architecture which performs three propagation steps during the forward pass and convolves 3rd order neighborhood of every node. We take the dimension of the output layer of the first and second layers as 256 and 128. For the decoder, we use a simple inner product decoding scheme.

All experiments were carried out on Linux server having 50 cores and *x*86_64 platform.

To validate the clustering results, we utilized two performance metrics, Adjusted Rand Index (ARI) and Silhouette Score [45]. Table 2 depicts the efficacy of *sc-CGconv* over the other methods. For the other competing methods, we select the top 1000 features/genes in the gene selection step and use the default clustering technique meant for this method. For sc-CGconv, after obtaining the low dimensional embedding of 128-dimension, we performed a simple k-means for clustering the cells. It is evident from the Table that sc-CGconv provides higher ARI (and average silhouette width) values for all four datasets. To know the effectiveness of the feature extraction method within the sc-GCconv, we have replaced it with PCA while keeping the

**Table 2. Comparison with state-of-the-art: Adjusted Rand Index (ARI) and Average Silhouette Width (ASW) are reported for seven competing methods on four datasets.**

| Dataset | Method | | | | | | | | | |
|---|---|---|---|---|---|---|---|---|---|---|
| | sc-CGconv | | Gini Clust | | GLM-PCA+Kmeans | | Fano+Kmeans | | Seurat | |
| | ARI | ASW | ARI | ASW | ARI | ASW | ARI | ASW | ARI | ASW |
| Baron [32] | 0.68 | 0.52 | 0.6 | 0.48 | 0.42 | 0.4 | 0.52 | 0.46 | 0.62 | 0.47 |
| Melanoma [34] | 0.43 | 0.45 | 0.56 | 0.52 | 0.15 | 0.29 | 0.18 | 0.24 | 0.42 | 0.29 |
| Klein [33] | 0.86 | 0.8 | 0.76 | 0.7 | 0.43 | 0.58 | 0.4 | 0.3 | 0.8 | 0.72 |
| PBMC [1] | 0.50 | 0.3 | 0.51 | 0.46 | 0.38 | 0.29 | 0.31 | 0.26 | 0.29 | 0.14 |
| | scGeneFit+Kmeans | | SC3 | | M3drop | | sc-GCconv (PCA) | | | |
| | ARI | ASW | ARI | ASW | ARI | ASW | ARI | ASW | | |
| Baron [32] | 0.62 | 0.43 | 0.60 | 0.4 | 0.54 | 0.48 | 0.60 | 0.49 | | |
| Melanoma [34] | 0.25 | 0.4 | 0.38 | 0.35 | 0.33 | 0.26 | 0.38 | 0.34 | | |
| Klein [33] | 0.82 | 0.75 | 0.80 | 0.66 | 0.67 | 0.54 | 0.71 | 0.76 | | |
| PBMC [1] | 0.47 | 0.48 | 0.48 | 0.31 | 0.35 | 0.3 | 0.41 | 0.30 | | |

other procedure intact. The last column of Table 2 shows the results of sc-Gcconv with PCA as feature extractor.

## sc-CGconv preserves cell-to-cell variability

Once features or low dimensional embedding are estimated to be important, it is essential to ask whether the cell-to-cell variability has been preserved within this low dimensional space. To determine this, we computed the Euclidean distance between each pair of cells, both in original dimension and in low dimension space. Thus two Euclidean distance matrices are obtained, one for the original feature space, and the other for the reduced dimension. The Correlation score (Kendall $\tau$) is computed between these two distance matrices. Fig 3 depicts the correlation measures for all the competitive methods in the four scRNA-seq datasets.

## sc-CGconv can identify marker genes

We followed the conventional procedure of Scanpy to find out markers (DE genes) from the clustering results. The clustering is performed on the low dimensional embedding obtained from the sc-GCconv. Scanpy utilized the Leiden graph clustering method to group the cells into different clusters. Wilcoxon rank-sum test [46] is utilized to find out the significant ($p < 0.05$) DE genes for each cluster which are treated as marker genes. We took the top 50 marker genes with their p-value threshold of 0.05 on the PBMC $68k$ dataset.

We found that 19 marker genes from the Melanoma dataset, 12 marker genes from the PBMC dataset, 8 marker genes from the Baron dataset, and 4 marker genes from the Klein dataset, are biologically significant according to CellMarker database [47]. The list of biologically significant marker genes for all four datasets is given in Table A in S1 File. The results of marker gene analysis of the Melanoma data can be found in S2 Text, Table A, and Figs B and C in S1 File.

## Execution time

All experiments were carried out on a Linux server having 50 cores and *X86_64* platform. As our proposed method is a deep learning based feature extraction method, it takes more time than any filter-based feature selection technique (e.g. *Fano*, *Gini − clust*). To check how the competing methods scale with the number of cells (and classes) we performed an analysis. We

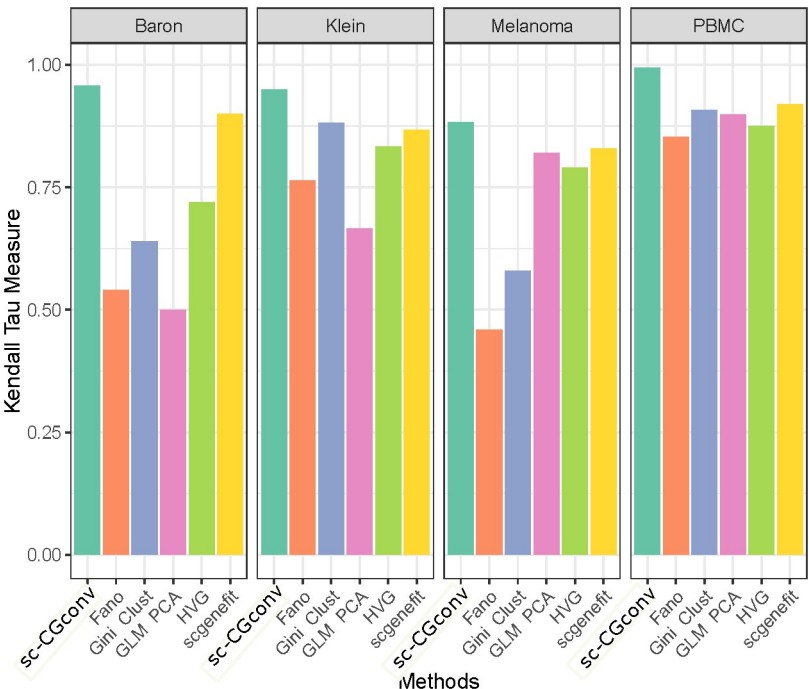

**Fig 3. Correlation score between two distance matrices, defined on original and reduced dimension.** Figure shows the comparisons among the competing methods based on the correlation scores (Kendall $\tau$) obtained from four different scRNA-seq datasets.

**Table 3. Execution time in minute for eight competing methods.**

| Datasets | # Cells | # Class | Execution Time (in Minute) | | | | | | | |
|---|---|---|---|---|---|---|---|---|---|---|
| | | | sc-CGconv | Gini Clust | GLM-PCA | Fano | Seurat | scGeneFit | SC3 | M3drop |
| Data1 | 500 | 2 | 9 | 2 | 1 | 1 | 3 | 3 | 5 | 4 |
| Data2 | 1000 | 3 | 13 | 2 | 1 | 1 | 7 | 5 | 8 | 6 |
| Data3 | 1500 | 4 | 17 | 3 | 1 | 3 | 11 | 10 | 13 | 12 |
| Data4 | 2000 | 5 | 20 | 5 | 3 | 5 | 14 | 13 | 17 | 15 |

have generated simulated data (using splatter) by varying the number of cells (and classes). Four simulated data are generated with the number of cells and classes as follows: 500 cells with two classes, 1000 cells with three classes, 1500 cells with four classes, and 2000 cells with five classes. All data are generated with equal group probabilities, 2000 number of features, fixed dropout rate (0.2), and 40% DE gene proportion. 1000 features are selected in each compared method and 128 embedded features are chosen using sc-CGconv. The runtime is compared with all seven different competing methods. The execution time (minute) for each dataset is given in Table 3.

## Conclusions

In this paper, we have developed a step-wise sampling based feature extraction method for scRNA-seq data leveraging the Copula dependency measure with graph convolution network. On one hand, LSH based sampling is used to deal with ultra-large sample size, whereas Copula dependency is utilized to model the interdependence between features (genes) to construct the

cell-cell graph. Graph convolution network has been utilized to learn low dimensional embedding of the constructed graph. There are four striking characteristics of the proposed method: I) It can sample a subset of features from original data keeping the structure intact. Therefore, minor clusters are not ignored. This sampling is achieved by using the LSH based sampling method. II) sc-CGconv utilizes scale invariant dependency measure which gives a superior and stable measure for constructing the dependency graph among the cells. III) GCN provides topology-preserving low dimensional embedding of the cell graphs. It can effectively capture higher-order relations among cells. IV) LSH based structure-aware sampling of features showed a significant lift in the accuracy (Correlation, ARI values) in large single cell RNA-seq datasets.

Another important observation is that sc-CGconv yields the highest ARI values for Klein and Pollen datasets in comparison to other State-Of-The-Art methods. The rationale behind this is that sc-CGconv utilized copula correlation measure, which correctly models the correlations among the feature set. In the holistic viewpoint, the sc-CGconv algorithm performs much better than the other methods.

The computation time of sc-CGconv is equivalent to the number of sampled features. The process may be computationally expensive when a large number of features are selected in the LSH step. However, as copula correlation returns stable and non-redundant features, in reality, a small set of selected features will be effective to construct the cell-cell graph. We observed in scRNA-seq data 1000 sampled features will serve the purpose.

Taken together, the proposed method *sc-CGconv* not only outperforms in topology preserving generation of cell embedding but also can able to identify good clusters for large single cell data. It can be observed from the results that *sc-CGconv* leads both in the domain of single cell clustering by extracting informative features and generating low dimensional embedding of cells from large scRNA-seq data. The results prove that *sc-CGconv* may be treated as an important tool for computational biologists to investigate the primary steps of downstream analysis of scRNA-seq data.

## Method

### Overview of datasets

We used four public single-cell RNA sequence datasets downloaded from Gene Expression Omnibus (GEO) https://www.ncbi.nlm.nih.gov/geo/ and 10X genomics (https://support.10xgenomics.com/single-cell-gene-expression/datasets). Table 4 shows a summary of the used datasets. See S1 Text in S1 File for a detailed description of the datasets.

### The formal details of sc-CGconv

**Copula.** The term *Copula* [48] originated from a Latin word *Copulare*, which means 'join together'. The Copula is utilized in several domains in statistics to obtain joint distributions from uniform marginal distributions. Following the famous *Sklar's* theorem, Copula (*C*) function is defined as follows [49, 50]

**Table 4. A brief summary of the dataset used here.**

| Dataset | Dataset Description | #Features | #Instances | #Class |
|---|---|---|---|---|
| Baron [32] | Human pancreas cell | 20125 | 8569 | 8 |
| Klein [33] | Mouse Embryo Cell | 24175 | 2717 | 4 |
| Melanoma [34] | Human Tumor Cell | 19783 | 68579 | 14 |
| PBMC68k [1] | Human Blood tissue | 32738 | 68793 | 11 |

Let, $(U_1, U_2, \cdots, U_n)$ be the random variables whose marginal distributions are uniform over [0, 1]. A copula function $C : [0, 1]^n \rightarrow [0, 1]$ is defined as the joint distribution:

$$C(u_1, u_2, \cdots, u_n) = P(U_1 \leq u_1, U_2 \leq u_2, \ldots, U_n \leq u_n). \tag{3}$$

Sklar's theorem extends this definition to more general random variables with possibly non-uniform marginals. The theorem states that, for any set of $n$ random variables $(X_1, \ldots, X_n)$, their joint cumulative distribution function $H(x_1, \cdots, x_n) = P[X_1 \leq x_1 \ldots X_n \leq x_n]$ can be described in terms of its marginals $F_i(x_i) = P[X_i \leq x_i]$ and a Copula function $C$, formally stated as:

$$H(x_1, x_2, \cdots, x_n) = C(F_1(x_1), F_2(x_2), \cdots, F_n(x_n)). \tag{4}$$

Among several categories of Copulas [51], Clayton Copula from Archimedean family is one of the most widely used function for high dimensional datasets [52].

**Clayton Copula.** Let, $\phi$ be a strictly decreasing function such that $\phi(1) = 0$, and $\phi^{[-1]}(x)$ is the pseudo inverse of $\phi(t)$ such that $\phi^{[-1]}(x) = \phi^{-1}(x)$ for $x \in [0, \phi(0))$ and $\phi^{[-1]}(t) = 0$ for $x \geq \phi(0)$. Let $U_1, U_2, \ldots, U_n$ be the random variables having uniform marginal distributions. Then, the general family of Archimedean copula is described as,

$$\begin{aligned} &C_{Archi}(u_1, u_2, \cdots, u_n) \\ &= \phi^{[-1]}(\phi(u_1) + \phi(u_2), \cdots, + \phi(u_n)), \end{aligned} \tag{5}$$

where, $\phi(.)$ is called the generator function. The Clayton Copula is a particular Archimedian copula when the generator function $\phi$ is given by,

$$\phi(x) = \frac{(x^{-\theta} - 1)}{\theta}, \tag{6}$$

with $\theta \in [-1, \infty)0)$.

**Copula based correlation measure (Ccor).** We model the dependence between two random variables using Kendall tau$(\tau)$ [53] measure. Note that we defined Kendall tau correlation using copula based dependence measure. In [54], Ding et al. first proposed a way to define the correlation by using copula dependence measure. They proposed *Ccor* as a robust and equitable measure which is defined as half the $L1$-distance of the copula density function from independence. In [13, 55]*Ccor* was defined in terms of Kendall's tau $(\tau)$ measure. Here we retain the definition of *Ccor* proposed in [13]. Kendall's tau $(\tau)$ is the measure of concordance between two variables; defined as the probability of concordance minus the probability of discordance. Formally this can be expressed as

$$\tau_{XY} = [P(x_1 - x_2)(y_1 - y_2) \geq 0] - [P(x_1 - x_2)(y_1 - y_2) \leq 0] \tag{7}$$

The concordance function $(Q)$ is the difference of the probabilities between concordance and discordance between two vectors $(x_1, y_1)$ and $(x_2, y_2)$ of continuous random variables with different joint distribution $H1$ and $H2$ and common margins $F_X$ and $F_Y$. It can be proved that the function $Q$ depends on the distribution of $(x_1, y_1)$ and $(x_2, y_2)$ only through their copulas. According to Nelson [48], there is a relation between Copula and Kendall $\tau$ that can be

expressed as:

$$\tau_{C(X,Y)} = \tau_{XY} = 4 \int\limits_{0}^{+1}\!\!\int C(u,v)\, dC(u,v) - 1, \tag{8}$$

Where, $u \in F_X(x)$ and $v \in F_Y(y)$. $\tau(C_{X,Y})$ is termed as *Ccor* in our study. Here the copula density $C(u, v)$ is estimated through the clayton copula defined in the previous section.

We have used $\tau_{C(X,Y)}$ to model the dependency between transcriptomic profiles among the cells.

## Feature extraction using sc-CGconv

sc-CGconv takes a stepwise approach for feature extraction from the scRNA-seq data: first, it obtains a sub-sample of genes using locality sensitive hashing, next it generates a cell neighborhood graph by utilizing the copula correlation (*Ccor*) measure, and finally, a graph representation learning algorithm (here GCN) is utilized to get the low dimensional embedding of the constructed graph.

**Structure preserving feature sampling using LSH.** LSH [28] reduces the dimensionality of higher dimension datasets using an approximate nearest neighbor approach. LSH uses a random hyperplane based hash function, which maps similar objects into the same bucket. LSH is used to partition the data points (genes) of the preprocessed count matrix ($M'$) into $k$ (here $k = 10$) different buckets such that $|G'| > 2^k$, where $G' = \{g'_j, j = 1, \cdots, ge'\}$ is the set of genes in $M'$. A $k$-nn graph is formed by searching the five nearest neighbours within the bucket for each gene. Each gene is *visited* sequentially in the same order as it appears in the original dataset and is added to the *selected* list while discarding its nearest neighbours. If the visited gene is discarded previously, then it will be skipped and its neighbors will be discarded. Thus a sub-sample of genes is obtained, which is further down-sampled by performing the same procedure recursively. The number of iterations for downsampling is user defined and generally depends on the size of the data points. We use cosine distance to compute the nearest neighbours of a gene. LSHForest [56] python package is utilized to implement the whole process.

Thus, a subset of $m$ number of genes, where,($m < ge'$) are obtained from the above sampling stage. These genes are considered as feature set, $F_{LSH} = \{f_s : s = 1, \cdots, m\}$ for the next stage of cell-cell graph construction.

**Cell neighbourhood graph construction.** For graph construction, we rank each node (cell) according to the *Ccor* values. For a node (cell) we compute the *Ccor* values to all of its possible pairs. A k-nearest neighbour list is prepared for each node based on the *Ccor* values. A high value of *Ccor* assumes there is a high similarity between the cell pair over the transcriptomic profile, while a smaller value signifies low similarity. The output of this step is an adjacency matrix representing the connection among the cells according to the k-nearest neighbour list and a node feature matrix storing the *Ccor* values for each node pair.

**Extracting node features using GCN.** We have utilized graph convolution network (GCN) [30] to learn the low dimensional embedding of nodes from the cell-cell graph. Given a graph $G = (V, E)$, the goal is to learn a function of signals/features on $G$ which takes i) A optional feature matrix $X \in N \times D$, where $x_i$ is a feature description for every node $i$, $N$ is the number of nodes and $D$ is the number of input features and ii) A description of the graph structure in matrix form which typically represents adjacency matrix $A$ as inputs, and produces a node-level output $Z \in N \times F$, where $F$ represents the output dimension of each node feature. The graph-level outputs are modeled by using indexing operation, analogous to the pooling operation uses in standard convolutional neural networks [57].

In general every layer of neural network can be described as a non-linear function:

$$H^{(l+1)} = f(H^{(l)}, A),$$ (9)

where $H^{(0)} = X$ and $H^{(l)} = Z$, $l$ representing the number of layers, $f(.,.)$ is a non linear activation function like *ReLU*. Following the definition of layer-wise propagation rule proposed in [30] the function can be written as

$$f(H^{(l)}, A) = \sigma(\hat{D}^{-1/2}\hat{A}\hat{D}^{1/2}H^{(l)}W^{(l)}),$$ (10)

where $\hat{A} = A + I$, $I$ represents identity matrix, $\hat{D}$ is the diagonal node degree matrix of $\hat{A}$, $\hat{D}_{ii} = \sum_j A_{ij}$, $W$ represents trainable weight matrix of the neural network. Intuitively, the graph convolution operator calculates the new feature of a node by computing the weighted average of the node attribute of the node itself and its neighbours. The operation ensures identical embedding of two nodes if the nodes have identical neighboring structures and node features. We adopted the GCN architecture similar to [30], a 3-layer GCN architecture with randomly initialized weights. For the cell-cell graph, we take the adjacency matrix ($A$) of the neighbourhood graph and put identity matrix ($I$) as the node feature matrix. The 3-layer GCN performs three propagation steps during the forward pass and effectively convolves the 3rd-order neighborhood of every node. The encoder uses a graph convolution neural network (GCN) on the entire graph to create the latent representation

$$Z = GCN(A, X)$$ (11)

The encoder works on the full adjacency matrix $A \in R^{n \times n}$ and $X \in R^{n \times m}$ is the node feature matrix, obtained from LSH step. Here we used a simple inner product decoder that try to reconstruct the main adjacency matrix $A$. The decoder function follows the following equation:

$$A(i, j) = \text{Sigmoid}(z_i^T \cdot z_j), \ \forall (i, j) \in V \times V$$ (12)

where $z_i, z_j$ reflect the representations of nodes $i, j$, as computed by the encoder. The trained model is applied to the test edges to see how effectively it can discover the deleted edges (see 'Training of GCN on cell-cell graph' in Result section). After training and evaluation of the model, the low dimensional embedding is kept and used in the cell clustering task.

## Supporting information

**S1 File. Supporting information.** S1 Text: Overview of the datasets. S2 Text: Marker analysis with sc-CGconv. Table A: Marker genes identified from the clustering results with sc-CGconv. Fig A: Performance of GCN on networks created from four datasets: receiver operating characteristic (ROC) curve for the validation is given for four datasets (see Table 1 of the main text for ROC score). Fig B: Marker analysis using sc-CGconv. After clustering DE genes are identified using clustering of PBMC data with sc-CGconv and results of marker genes on ultra large PBMC datasets. Panel-A. 2D UMAP visualization of PBMC dataset with original cell annotations. Panel-B. 2D UMAP visualization of clustering results with sc-CGconv. Panel-C. visualization of 12 markers which are overlayed based on their expression –low (blue) to high (yellow)- on the reference PBMC UMAP plot. Fig C: Figure depicts the results of marker gene analysis on melanoma datasets. Panel-A. 2D UMAP visualization of melanoma data with original annotation. Panel B. 2D UMAP visualization of clustering results with sc-CGconv. Panel-C. visualization of 9 markers which are overlayed based on their expression –low (blue) to high (yellow).
(PDF)

## Author Contributions

**Conceptualization:** Sumanta Ray.

**Data curation:** Snehalika Lall.

**Formal analysis:** Snehalika Lall, Sumanta Ray.

**Funding acquisition:** Sanghamitra Bandyopadhyay.

**Investigation:** Snehalika Lall, Sumanta Ray, Sanghamitra Bandyopadhyay.

**Methodology:** Snehalika Lall, Sumanta Ray, Sanghamitra Bandyopadhyay.

**Project administration:** Sumanta Ray, Sanghamitra Bandyopadhyay.

**Resources:** Snehalika Lall.

**Software:** Snehalika Lall, Sumanta Ray.

**Supervision:** Sumanta Ray, Sanghamitra Bandyopadhyay.

**Validation:** Snehalika Lall, Sumanta Ray, Sanghamitra Bandyopadhyay.

**Visualization:** Snehalika Lall, Sumanta Ray.

**Writing – original draft:** Snehalika Lall, Sumanta Ray.

**Writing – review & editing:** Snehalika Lall, Sumanta Ray, Sanghamitra Bandyopadhyay.

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
