## [Decision Letter · Decision Letter 0]

11 Dec 2021

Dear Dr. Ray,

Thank you very much for submitting your manuscript "A copula based topology preserving graph convolution network for clustering of single-cell RNA-seq data" for consideration at PLOS Computational Biology.

Reviewers have now commented on your paper. You will see that they are advising that you revise your manuscript. If you are prepared to undertake the work required, I would be pleased to reconsider my decision.

We cannot make any decision about publication until we have seen the revised manuscript and your response to the reviewers' comments. Your revised manuscript is also likely to be sent to reviewers for further evaluation.

Sincerely,

Quan Zou

Guest Editor

PLOS Computational Biology

Ilya Ioshikhes

Deputy Editor

PLOS Computational Biology

Reviewers have now commented on your paper. You will see that they are advising that you revise your manuscript. If you are prepared to undertake the work required, I would be pleased to reconsider my decision.

Reviewer's Responses to Questions

**Comments to the Authors:**

Reviewer #1: In this study, Lall et al introduce sc-CGconv (Copula based graph convolution network for single cell clustering). This is an unsupervised feature extraction and clustering approach that uses copula correlation (Ccor) followed by graph convolution based clustering approach r to extract biologically relevant gene clusters obtained from single cell RNASeq data. The authors compare their method with 5 other popular methods of gene selection and clustering and discuss the advantages of their method over other popular methods for scRNA seq cell clustering. Overall, the manuscript is well written and provides a new method for understanding biologically meaningful signals arising out of scRNA seq data.

The manuscript could be considered fit for publication once the following issues are addressed:

1. Line 35: “The standard approaches for gene selection (or feature extraction) are failed..” Consider rephrasing to: “The standard approaches for gene selection (or feature selection) fail to…”

2. Line 37: “Moreover, the exiting approaches…” Typo here. “Moreover, existing approaches...”

3. Line 56: “The advantages of sc-CGconv come selecting a new robust equitable …” Consider substituting ‘come’ or use another appropriate phrase.”

4. Figure 1: Capitalize ‘W’ in figure legend. Also a typo is observed: “adjacencey matrix”.

5. Line 70: Why is ‘experiments’ italicized?

6. Line 71: “(ii) the annotation of cells are accurate ...” Replace “are” with “is”

7. Lines 81 and 96: It is seen that 81 and 96 state “See -A of figure 1” and “See C in Figure 1” whereas Lines 88 and 103 state “See panel-B” and “See panel-D”. Please consider mentioning either panel-A or Figure 1A,1B (whichever is appropriate).

8. Line 84: Consider rephrasing. “… a thousand genes expression...” This sentence could be written as “We choose cells having more than 1000 genes with non-zero expression values…” Also, the authors could provide an explanation of how the structure of the data matrix is preserved. How does LSH achieve that?

9. Line 96: “A robust equitable correlation measure Ccor”. The manuscript would greatly benefit if the authors provided an overview of the copula correlation concept to a greater extent. How does this method compare to other correlation metrics? A brief literature survey about the usage of this concept is also warranted to drive home the usage of this correlation measure. Which other fields use copula-based correlation and to what extent are they helpful?

10. Line 111: “The result of this step is a feature matrices (Fi)..” Grammatical error – “… is a feature matrix (Fi)..”

11. Line 112: Which network features are inferred here? Unclear.

12. Line 120: Consider providing an explanation behind the use of Adam optimizer and ReLu as activation function. A comment on the learning rate for the optimization procedure would be helpful.

13. Line 122: Manuscript could benefit by the addition of a figure for the ROC curve, in addition to the ROC statistics in table1.

14. Table1: Table legend states –“First two columns of the table shows total number of nodes and number of edges..” whereas in the table itself number of edges is written first followed by number of nodes.

15. Line 125: Wolf et al should be capitalized. Manuscript states “…wolf et al”.

16. Line 126: Consider writing the full name of Kullback-Leibler (KL) divergence since this is the first time

17. Line 129: “We measured the KL divergenece”. Typo here.

18. Line 132: The correct usage of the phrase is “state of the art” and not “arts”. Please rectify. Also in Table 2 (Table caption).

19. Figure 2: ‘KL div’ as a short form for KL divergence should be mentioned in the legend.

20. Line 157: What is the mathematical interpretation of the Clayton Copula? How does the value of theta matter? What is its physical implication towards the performance of scCGconv?

21. Line 174: “Once features / (low dimensional embedding) …” Please consider replacing / with or and remove parentheses. It is confusing.

22. Line 179 and Figure 3 (Legend): What do the authors mean by “..(Kandle tau “ in line 179 and “Kandel Tau in Figure 3 legend..”? Do the authors intend to say Kendall tau? Please rectify if this is the case.

23. Line 180: grammatical error - “…for all competitive method…” should be - “…for all competitive methods…”

24. Line 182: This section should be considered for rewriting. How exactly are the authors identifying marker genes using their method? A detailed explanation is warranted here.

25. Also line 187 states 19 markers from melanoma and 13 from PBMC dataset are significant; whereas table 3 states 19 and 12. Which numbers are correct?

26. Line 188: If cell marker is the name of the database, please capitalize the name as in Cell Marker?

27. Table 3: What do the authors mean by pubmed id of the markers? The numbers in parenthesis do not refer to the NCBI gene id or Ensembl Gene Stable ID. Please provide an explanation of what nomenclature of genes is used here. Also why do all 3 CD8 T cell markers have the same ID i.e. 28622514 although they correspond to 3 different genes? Are these IDs specific to the Cell Marker database used by the authors?

28. Line 189: What are the markers identified by scCGconv for other 2 datasets (Baron and Klein)? It would be beneficial to have a list of these markers as a Supplementary Table.

29. Line 209: What is meant by Human Klein and Pollen? Please explain.

30. In general, how does this method compare to other NN utilizing scRNAseq methods like PHATE and SAUCIE? Please comment on this.

31. It would be good to see how the list of features selected by this method compares to lists of genes selected by other methods (for example methods that use standardized variance / Fano or Gini index)

32. It would also be useful to see an exploration of the properties of genes. For example, Gini picks up genes expressed in few cells, Fano picks up genes with large variance with respect to the mean expression. What are the variational properties of the genes picked up by their method?

33. The authors don't do any intermediate dimensionality reduction (typically one would do feature selection, run PCA, and use the PCs for further computation rather than genes). The construction of the cell kNN graph is done directly in the selected feature space. This could be justified since their gene selection is supposed to remove redundant information, so running PCA might not add much. But this is another place where a formal comparison of the two approaches would be worthwhile to see. PCA and their gene selection method are both essentially trying to remove redundant information; so if one compares their approach with one where there is no feature selection and only PCA (and using PCs for further steps), what does the comparison look like? One example comparison: are the genes that contribute heavily to top PCs in PCA the same (or similar) as the genes retained by their feature selection method?

Reviewer #2: In this manuscript, Lall et al. proposed a novel feature selection and clustering approach named sc-CGconv for single-cell RNA-seq data. sc-CGconv first performs LSH based sampling to obtain structure-aware features and then constructs cell-cell graph using copula correlation. Finally, low dimensional embedding is extracted from graph convolution network. Overall, Sc-CGconv shows great performance in comparison with state-of-the-art clustering methods, it may be useful for researcher in this field. I would like to see the authors address the following comments.

1. M3Drop is an important and popular feature selection method tailed for scRNA-seq data, the authors also mentioned it for several times in the manuscript. However, there is no comparison with it in Table 2. I would suggest the authors to add it to get a more comprehensive comparison.

2. As we know, deep learning is a powerful tool, but it often tends to consume a lot of time. To let users fully understand the characteristics of the proposed approach, the authors should provide the running time of all the methods used in this study for all four datasets.

3. After I successfully installed the R package “copulaGCN” following the authors’ instructions at Github (https://github.com/Snehalikalall/CopulaGCN), running “library(copulagcn)” still got an error. The correct name of the package may be “ucfs”? Please check the package name and modify the correct usage in the README.

4. I believe all feature selection methods could easily identify several marker genes, therefore it is not necessary to highlight Table 3. I would urge the authors to put Table 3 in the supplementary files.

5. In the legend A of Fig1, method “limnorm” does not exist. The authors may refer to “linnorm”.

6. Several important reviews concerning feature selection and clustering of scRNA-seq data are missing, such as:

1) Zhang Z, Cui F, Wang C, Zhao L, Zou Q: Goals and approaches for each processing step for single-cell RNA sequencing data. Briefings in bioinformatics 2021, 22(4).

2) Zhu X, Li H-D, Guo L, Wu F-X, Wang J: Analysis of Single-Cell RNA-seq Data by Clustering Approaches. Current Bioinformatics 2019, 14(4):314-322.

**Have the authors made all data and (if applicable) computational code underlying the findings in their manuscript fully available?**

Reviewer #1: Yes

Reviewer #2: None

PLOS authors have the option to publish the peer review history of their article (what does this mean?). If published, this will include your full peer review and any attached files.

Reviewer #1: No

Reviewer #2: No
---

## [Decision Letter · Decision Letter 1]

27 Jan 2022

Dear Dr. Ray,

We are pleased to inform you that your manuscript 'A copula based topology preserving graph convolution network for clustering of single-cell RNA-seq data' has been provisionally accepted for publication in PLOS Computational Biology.

Best regards,

Quan Zou

Guest Editor

PLOS Computational Biology

Ilya Ioshikhes

Deputy Editor

PLOS Computational Biology

Reviewer's Responses to Questions

**Comments to the Authors:**

Reviewer #1: The questions and concerns raised in the review process were satisfactorily addressed. The research was performed with rigor and the results presented were easy to interpret. I recommend that the manuscript should be accepted for publication.

Reviewer #2: I am satisfied with the answers the authors provided to my questions and the revisions they added to the manuscript and supplementary information.

**Have the authors made all data and (if applicable) computational code underlying the findings in their manuscript fully available?**

Reviewer #1: Yes

Reviewer #2: Yes

PLOS authors have the option to publish the peer review history of their article (what does this mean?). If published, this will include your full peer review and any attached files.

Reviewer #1: No

Reviewer #2: No

---

## [Editor Report · Acceptance letter]

7 Mar 2022

PCOMPBIOL-D-21-01961R1 

A copula based topology preserving graph convolution network for clustering of single-cell RNA-seq data

Dear Dr Ray,

I am pleased to inform you that your manuscript has been formally accepted for publication in PLOS Computational Biology. Your manuscript is now with our production department and you will be notified of the publication date in due course.

With kind regards,

Anita Estes
